# Effects of Bioactive Marine-Derived Liposomes on Two Human Breast Cancer Cell Lines

**DOI:** 10.3390/md18040211

**Published:** 2020-04-13

**Authors:** Jie Li, Kamil Elkhoury, Claire Barbieux, Michel Linder, Stéphanie Grandemange, Ali Tamayol, Grégory Francius, Elmira Arab-Tehrany

**Affiliations:** 1CRAN, CNRS-Université de Lorraine, F-54506 Vandœuvre-lès-Nancy, France; lijiewhu@126.com (J.L.); claire.barbieux@inserm.fr (C.B.); stephanie.grandemange@univ-lorraine.fr (S.G.); 2LIBio, Université de Lorraine, F-54000 Nancy, France; kamil.elkhoury@univ-lorraine.fr (K.E.); michel.linder@univ-lorraine.fr (M.L.); 3Department of Biomedical Engineering, University of Connecticut, Mansfield, CT 06269, USA; atamayol@uchc.edu; 4LCPME, CNRS-Université de Lorraine, F-54600 Villers-lès-Nancy, France; gregory.francius@univ-lorraine.fr

**Keywords:** PUFA, omega-3, breast cancer, liposomes, DHA, EPA

## Abstract

Breast cancer is the leading cause of death from cancer among women. Higher consumption of dietary marine n-3 long-chain polyunsaturated fatty acids (LC-PUFAs) is associated with a lower risk of breast cancer. Eicosapentaenoic acid (EPA) and docosahexaenoic acid (DHA) are two n-3 LC-PUFAs found in fish and exert anticancer effects. In this study, natural marine-derived lecithin that is rich in various polyunsaturated fatty acids (PUFAs) was extracted from salmon heads and transformed into nanoliposomes. These nanoliposomes were characterized and cultured with two breast cancer lines (MCF-7 and MDA-MB-231). The nanoliposomes decreased the proliferation and the stiffness of both cancer cell types. These results suggest that marine-derived lecithin possesses anticancer properties, which may have an impact on developing new liposomal delivery strategies for breast cancer treatment.

## 1. Introduction

Female breast cancer is the most common cancer worldwide, with about 2.1 million newly diagnosed cases in 2018 [1]. Among women, breast cancer is not only the most diagnosed cancer but also the leading cause of cancer death [1]. Notably, a systematic review of studies published between 2007 and 2011 concluded that higher intakes of dietary and supplemental omega-3 long-chain polyunsaturated fatty acids (n-3 LC-PUFAs) are associated with a lower risk of breast cancer [2]. The same conclusion was reached by a meta-analysis of data from 21 independent prospective cohort studies [3]. Moreover, the association of n-3 LC-PUFAs and the incidence of several cancer types have been highlighted by numerous epidemiological studies [4,5,6,7,8]. In addition to that, n-3 LC-PUFAs intake is associated with positive effects on cardiovascular health [9,10,11], brain function [12,13], immunity [14], and inflammation [15].

The exact mechanisms involved in n-3 LC-PUFAs anticancer effects are poorly understood, but it has been reported that it is based on modulation of cellular proliferation and differentiation, alteration in gene expression, induction of apoptosis, generation of reactive oxygen species, lipid peroxidation, and increased drug transport across the cell membrane [16,17,18,19,20]. Lecithin extracted from salmon (*Salmo salar*) heads contains a high percentage of n-3 LC-PUFAs, such as docosahexaenoic acid (DHA) and eicosapentaenoic acid (EPA) [21]. It is believed that for the inhibition of the proliferation and invasion of breast cancer cells, DHA is more potent than EPA [22].

The cellular uptake efficiency of particles increases with decreasing particle size [23]. Nanoparticles in the range of 100 to 200 nm can avoid mechanical filtration by the spleen due to their small size but are also large enough to avoid selective uptake in the liver, and thus their bloodstream circulation time can be extended. Furthermore, due to their small size and a mechanism called the enhanced permeability and retention (EPR) effect, the passive targeting of tumor cells is possible, and their intracellular accumulation and localization in the tumor area are improved [24,25]. The ζ-potential is another important parameter, as negatively charged nanoparticles are not only significantly less phagocytized than positive ones but also possess improved physical stability [26].

Producing nanoliposomes from lecithin will not only improve their cellular uptake but will also allow for the encapsulation of hydrophobic bioactive agents that will improve their poor bioavailability. In addition, this encapsulation will increase the already existing therapeutic potential of nanoliposomes [27]. One such agent is curcumin, which showed improved anticancer effects when encapsulated in salmon-derived nanoliposomes [28]. Moreover, these natural nanoliposomes significantly increased the metabolic activity of cortical neurons [29,30].

Since the 1990s, it has been reported that when polyunsaturated fatty acids (PUFAs) content is increased in the diet of DBA/2 mice, some modifications in the functional and physical properties of the tumor cells’ membranes were caused through lipid accumulation, thus inducing a higher sensitivity to doxorubicin or hyperthermia treatment [31]. Moreover, tumor stiffening can hinder the efficacy of cancer drugs [32]. Therefore, nanoliposomes can form a deposit of lipids around the tumor area, and thus decrease anticancer drug resistance by softening the cells’ membranes.

Hence, the present study is focused on the production and multiscale characterization of negatively charged nanoliposomes from marine-derived lecithin rich in n-3 LC-PUFAs. More importantly, we report that these nanoliposomes can decrease the proliferation and the stiffness of two breast cancer lines (MCF-7 and MDA-MB-231), and thus can be used as potential anticancer drug delivery nanovesicles.

## 2. Results and Discussion

The total PUFAs (48.72%) were predominant in salmon lecithin with the presence of seven different types of PUFAs (Table 1). The most significant PUFAs were C22:6n3 (DHA), C20:5n3 (EPA), C18:2n6 (linoleic acid), and C18:3n3 (α-linolenic acid) with 30.12%, 9.54%, 5.79%, and 4.67%, respectively. The main fatty acid in the saturated class was C16:0 (12.09%) and in the monounsaturated class was C18:1n9 (20.20%). The ratio of n-3/n-6 was 5.10 and DHA/EPA was 3.16. The study of the lipid classes of salmon lecithin showed that phosphatidylcholine (42%) was a major class of phospholipids. The percentage of polar fraction in lecithin was 67.69% ± 0.9%, and the percentage of triacylglycerols (TAG) was 31.16% ± 0.4%. Additionally, salmon lecithin contained a small amount of cholesterol (1.14 ± 0.1%).

To facilitate and improve the cell uptake of salmon lecithin, nanoliposomes were generated and characterized (Figure 1). The average hydrodynamic diameter size of the produced nanoliposomes was determined using dynamic light scattering (DLS) and it was equal to 131 nm. The nanoliposomes presented a narrow distribution with a polydispersity index (PdI) of 0.29 and a negative ζ-potential of −54.8 mV. Nanoliposome particle size can be affected by the lipid composition and the chosen preparation method [33]. The small PdI (<0.3) shows that particles had a controlled size distribution and a narrow dispersity. The surface electrical charge of the particles is characterized by ζ-potential measurements [34]. The high value of the ζ-potential indicates that the nanoliposome dispersion presents good stability, as the greater the ζ-potential magnitude, the greater the repulsion between particles, which leads to a more stable colloidal dispersion [35].

The fluidity of the lipid membrane reflects the order and dynamics of phospholipid alkyl chains in the bilayer of vesicles [36]. The membrane fluidity level is tuned according to the membrane’s fatty acids (FAs) composition since the presence of saturated FAs increases the packing between phospholipids and thus reduces the membrane fluidity, whereas unsaturated FAs reduces this packing and increase the membrane fluidity [37]. Nanoliposomes derived from salmon lecithin had a relatively high membrane fluidity of 3.15 ± 0.07, which can be decreased by the encapsulation of a bioactive agent or by applying a polymer coating [38].

Our results performed on MCF-7 and MDA-MB-231 cells indicate that liposomes affect their proliferation rate. Indeed, crystal violet experiments (Figure 2) indicated that even at a low dose, liposomes can decrease significantly the level of attached cells after 72 h. These data were confirmed by the fact that this decrease is associated with an increase in the percentage of cells in the G0/G1 phase of the cell cycle (Figure 3). Thus, the decrease of attached cells observed with crystal violet staining is due to a decrease of cell proliferation without affecting cell viability since the Sub G1 phase, corresponding to dying cells, is not significantly increased after a long exposure with a high quantity of liposomes. Thus, our results indicate that negatively charged nanoliposomes from marine-derived lecithin can induce a cell cycle arrest in two breast cancer cell lines.

Modifications in mechanical properties of tumor cells subjected to liposomes treatment are related to different cellular processes associating liposome interactions with the cell membrane. AFM experiments were performed in order to quantify the mechanical properties of the cells before and after liposomes treatment at different incubation times, as presented in Figure 4 and Table 2. The statistical analysis of cell stiffness measured by AFM shows important variations when both MCF7 and MDA-MB231 cells were subjected to liposomes.

In Figure 4, MCF7 cell stiffness decreased by a factor of 2, from 5 to 3 kPa after 48 h treatment and up to 2 kPa after 96 h treatment. For MDA-MB231 cells, a similar phenomenon was observed. Indeed, the effect of liposomes was similar when comparing MDA-MB231 and MCF7 cells stiffness.

These results indicated that liposomes treatment led to cells softening and this decrease in stiffness may be related to the fusion of liposomes with tumor cells’ membranes. Such accumulation of phospholipids within the plasma membrane should probably disorganize it and decrease its cohesion/stiffness. Indeed, previous works highlighted that some lipids and proteins are able to regulate membrane rigidity when they are added to the latter [39,40,41].

In addition, some types of lecithin are known for their ability to reduce the interfacial surface tension of the lipid membrane [42,43]. Consequently, the softening of MDA and MCF7 cells may be due to the integration/fusion of nanoliposomes to their membrane, resulting in addition and accumulation of the salmon lecithin. 

These results are very interesting for further application in cancer therapy because several works evidenced that the softening of tumor cells can help in cancer treatment [44,45,46]. The abnormal stiffening of the tumor environment is associated with tumor development [47]. Through a local denaturation of adjacent tissues, nanomaterials can induce a remodeling and softening of the tumor microenvironment at various stages, which can lead to tumor regression [46]. So, the loss of stiffness of cancer cells represents a mechanical cue for tumor regression and enhances the delivery and thus the efficacy of chemotherapeutics [48]. Such mechanical modifications of tumor stiffness are usually performed using photothermal therapy and radiotherapy [46]. Here, we demonstrated that such mechanical modifications can be obtained using only salmon lecithin-based nanoliposomes.

## 3. Materials and Methods 

Salmon lecithin was obtained via enzymatic hydrolysis as previously described by Linder et al. [49]. Boron trifluoride (14% in methanol), acetonitrile (≥99.9%), chloroform (≥99.9%), methanol (≥99.9%), and hexane (≥99.9%) were all purchased from Fisher (Illkirch-Graffenstaden, France) and Sigma-Aldrich (Saint-Quentin Fallavier, France). Acetic acid (≥99.8%) was supplied by Prolabo-VWR.

### 3.1. Fatty Acid Composition

Fatty acid methyl esters (FAMEs) were prepared as previously described by Ackman [50]. The separation of FAMEs was carried out on a Shimadzu 2010 gas chromatography coupled with flame ionization detection (Perichrom, Saulx-lès-Chartreux, France). A fused silica capillary column was used (60 m, 0.2 mm i.d.× 0.25 µm film thicknesses, SPTM2380 Sopulco, Bellfonte, PA, USA). Injector and detector temperatures were set at 250 °C. For the first 3 min, the column temperature was set at 120 °C. Then, this temperature was increased to 180 °C at a rate of 2 °C/min and maintained at 220 °C for 25 min. PUFA1 and PUFA2 standard mixtures, obtained from marine and vegetable sources, respectively (Supelco, Sigma–Aldrich, Bellefonte, PA, USA), were used to identify fatty acids.

### 3.2. Lipid Classes

Lipidic classes of salmon lecithin were studied using an Iatroscan (MK-5 TLC-FID, Iatron Laboratories Inc., Tokyo, Japan) as previously described by Hasan et al. [28]. The proportion of polar and neutral lipid fractions were characterized using two migrations. Area percentages were shown as the average of three repetitions.

### 3.3. Nanoliposome Preparation

Two grams of lecithin were hydrated with 98 mL of distilled water and the suspension was agitated for 5 h under nitrogen. The samples were then sonicated at 40 kHz and 40% of full power for 300 s (1 s on, 1 s off) to obtain a homogeneous solution. Samples were stored in glass bottles in the dark at 4 °C.

### 3.4. Nanoliposome Size and ζ-Potential Measurements

Following the dilution of samples (1:200) with ultrapure water, their size, polydispersity index (PdI), and ζ-potential were measured via DLS (Zetasizer Nano ZS, Malvern, UK). Standard capillary electrophoresis cells were used to examine the nanoliposomes. Size and ζ-potential measurements were studied at 25 °C with an absorbance of 0.01, a fixed scattering angle of 173°, and a refractive index of 1.471.

### 3.5. Transmission Electron Microscopy (TEM)

The morphology of nanoliposomes was monitored via TEM using a negative staining method as previously described [30]. In brief, the liposomal formulation was diluted in distilled water (1:10 ratio) and then mixed with a 2% ammonium molybdate solution with a ratio of 1:1. The mixture was reserved at room temperature for 3 min. Then, one drop was placed and dried on a Formvar carbon-coated copper grid (200 mesh, 3 mm diameter HF 36). Then, the morphology of the nanoliposomes was examined using a Philips CM20 TEM equipped with an Olympus TEM CCD camera at 200 kV.

### 3.6. Membrane Fluidity

The membrane fluidity of nanoliposomes was measured by fluorescence anisotropy measurements as described by Maherani et al. [51]. In brief, TMA–DPH solution (1 mM in ethanol) was mixed with the liposomal suspension to reach a final concentration of 4 µM for the probe and 0.2 mg/mL for the lipid. Then, the fluorescent intensity was measured with Tescan INFINITE 200R PRO (Warriewood, Sydney, Australia) equipped with fluorescent polarizers. Samples were excited at 360 nm and emission was recorded at 430 nm under constant stirring at 25 °C. The Magellan 7 software was used for data analysis and to calculate the polarization value (P). Membrane fluidity was defined as 1/P.

### 3.7. Cell Culture

The two human breast cancer cell lines, MCF-7 and MDA-MB-231, were obtained from the European Collection of Cell Culture. Cells were cultured in RPMI 1640 medium without phenol red (Gibco™, ThermoFisher Scientific, Grand Island, NY, USA) supplemented with 10% (*v*/*v*) fetal bovine serum, 1% penicillin/streptomycin, and 2 mM L-Glutamine at 37 °C in a humidified atmosphere of 5% CO_2_.

### 3.8. Cell Viability Assay

The viability of cells was determined by crystal violet assay. MCF-7 and MDA-MB-231 were seeded in a 96 well-plates with a 100 µL medium at a density of 8 × 10^3^ per well and 1 × 10^4^ per well, respectively. After 24 h of culture, cells were treated for the indicated time with doses of liposomes. Then, cells were fixed and stained in 0.1% crystal violet for 30 min. After washing three times with distilled water, 100 µL of 10% acetic acid was added in each well and the plate was incubated on a bench rocker for 10 min. The absorbance was measured at 595 nm on a VICTOR Multilabel Plate Reader (PerkinElmer, Waltham, MA, USA). The absorbance of cells without treatment was set to 100%. Results are expressed as the percentage of absorbance values compared to the absorbance of untreated cells.

### 3.9. Cell Cycle Analysis

MCF-7 and MDA-MB-231 cells in the exponential phase of growth were treated with liposomes for the indicated time and doses. Then, cells were harvested by trypsinization, washed twice with ice-cold PBS, and fixed by 70% ethanol at −20 °C for at least 30 min. The cells were then incubated with 50 μg/mL of propidium iodide in the presence of 100 μg/mL RNase A for 30 min at room temperature. Cell cycles were analyzed using a BD FACSCalibur™ flow cytometer (BD Biosciences, Franklin Lakes, NJ, USA).

### 3.10. Nanomechanical Properties by Atomic Force Microscopy (AFM)

Imaging and nanoindentation measurements were performed using a Bioscope Resolve (Bruker Nano Surface, Bruker France SAS, Palaiseau, France) and an MFP3D-BIO instrument (Asylum Research Technology, Oxford Instruments Company, Wiesbaden, Germany) respectively. The topography of the cells was obtained by AFM operating in peakforce tapping™ mode. Silicon nitride cantilevers of conical shape purchased from Bruker (MLCT, Bruker France SAS, Palaiseau, France) with a spring constant of about 0.01 Nn nm^−1^ were used for both imaging and nanomechanical measurements. All images were recorded with a resolution of 512 × 512 pixels and a scan rate of 1 Hz. Nanomechanical properties of the cells were addressed in DMEM solution by recording at least 3 force–volume images (FVI) at different locations of the petri dish. Each FVI consisted of a grid of 50 × 50 force curves measured by adopting a 2 µm s^−1^ approach rate of the tip toward the sample. The cell stiffness (Young modulus) *E* was evaluated by analyzing the force–indentation curves within the framework of the Sneddon model. In this model, the Young modulus is related to the applied force according to the equation given below: (1)$$F=\frac{2E\cdot \mathrm{Tan}\left(\alpha \right)}{\pi \left(1-\nu^{2}\right)}R^{1/2}\delta^{2}\cdot f_{\mathrm{BECC}}$$
where *δ* is the indentation depth, *ν* is the Poisson coefficient, *α* is the semi-top angle of the conical tip, and *f*_BECC_ is the bottom effect cone-correction function that considers the stiffness of the petri dish substrate that supports the cells [52]. All FVI were analyzed using an automatic Matlab algorithm detailed elsewhere [53], and the average Young moduli values given in this work were derived from at least 800 force curves.

### 3.11. Statistical Analysis

All results are represented as mean value ± SEM. Statistical analyses were performed by using Student’s *t-*test, which compared untreated versus treated cells. Statistically significant results were represented as follows: * *p* < 0.05, ** *p* < 0.01, and *** *p* < 0.001.

## 4. Conclusions

In summary, we developed a natural biomimetic lecithin formulation rich in n-3 LC-PUFAs extracted from salmon heads. This formulation was used to produce liposomes, which were characterized to determine their morphological and physicochemical properties. The characterized nanosize and negative charge of the liposomal formulation will lead to improved cellular uptake and protection from macrophages. In addition, the effects of these nanoliposomes were assessed on MCF-7 and MDA-MB-231 breast cancer cell lines. Not only did these nanoliposomes decrease cancer cells’ proliferation by inducing a cell cycle arrest but they also decreased the cells’ stiffness and thus allowed for cancer cells softening. Since nanoliposomes can encapsulate and control the release of both hydrophilic and hydrophobic molecules, such mechanical softening will improve the limited anticancer drug penetration and release in tumor cells, and thus decrease the tumor’s resistance. These findings prove the potential of marine-derived nanoliposomal formulations as controlled drug delivery vehicles for breast cancer treatment.

## Figures and Tables

**Figure 1 marinedrugs-18-00211-f001:**
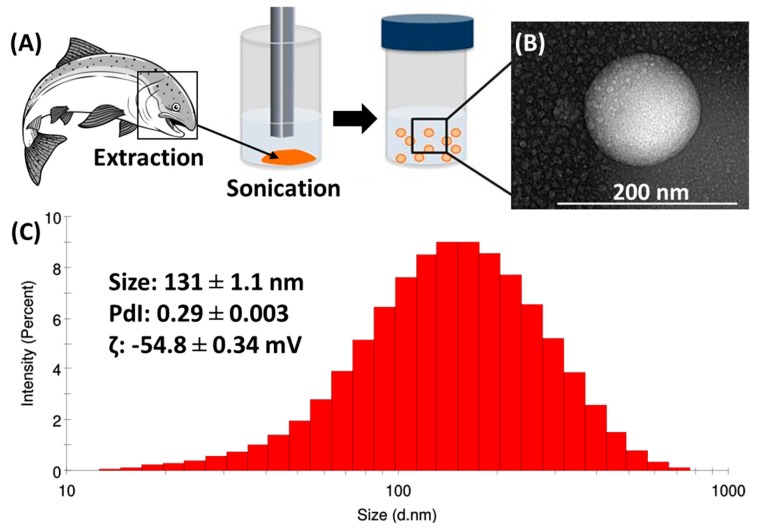
(**A**) Schematic of the extraction of lecithin from salmon head followed by its sonication to produce nanoliposomes. (**B**) The TEM image of the produced nanoliposomes. (**C**) The physicochemical characterization of the produced nanoliposomes.

**Figure 2 marinedrugs-18-00211-f002:**
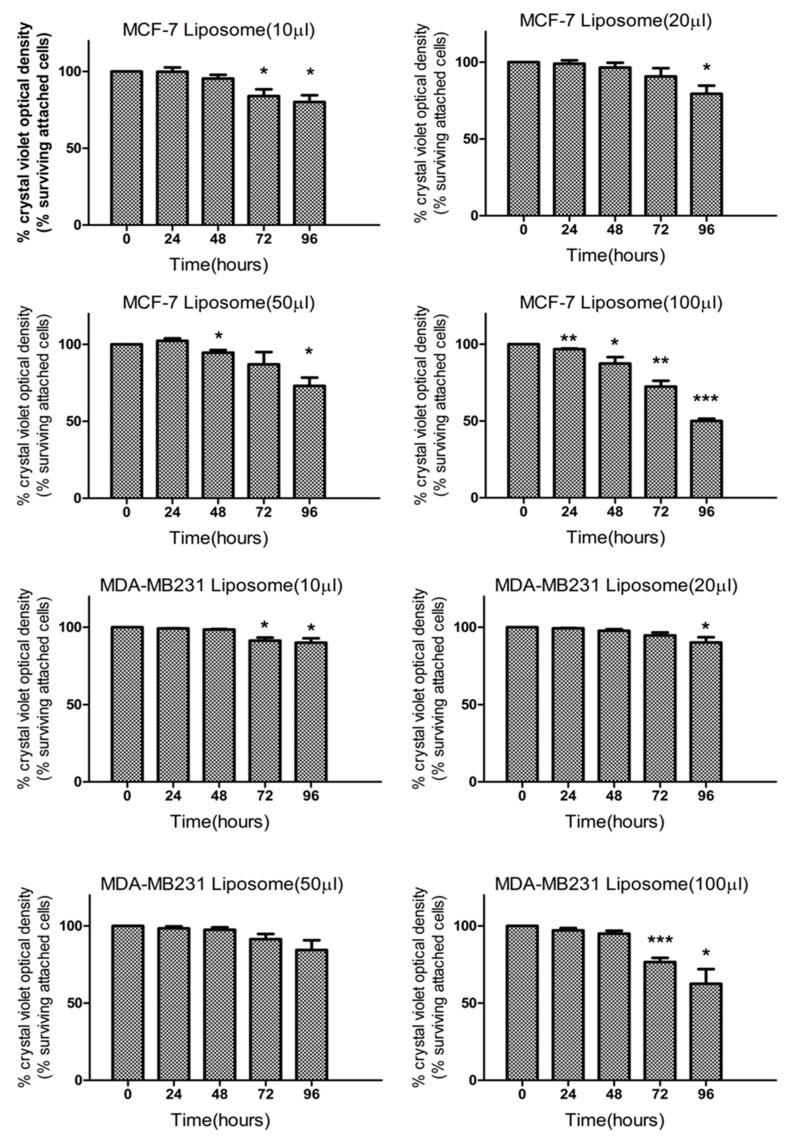
Effect of liposomes on MCF-7 and MDA-MB231 cell viability determined by crystal violet assay. After treatment with different doses of liposomes (10, 20, 50, or 100 µL of liposomes per 1 mL of culture media) for the indicated time (0, 24, 48, 72, and 96 h), cells were fixed and stained by 0.1% crystal violet for 30 min. After solubilization, absorbance was measured by a Perkin Elmer multilabel plate reader. Data are represented by the mean ± SEM of three independent experiments and are indicated as statistically significant for * *p* < 0.05, ** *p* < 0.01, and *** *p* < 0.001 based on a Student’s *t-*test.

**Figure 3 marinedrugs-18-00211-f003:**
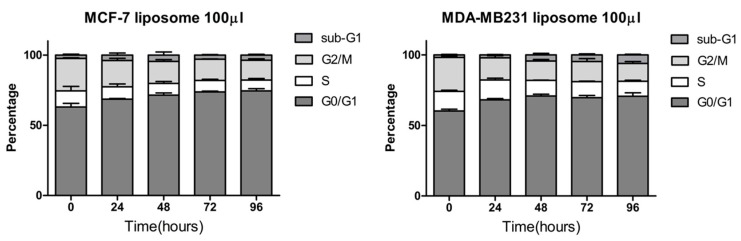
Cell cycle distribution of MCF-7 and MDA-MB231 cells after the addition of liposomes. Cells were treated with 100 µL of liposomes per 1 mL of culture media for the indicated time (0, 24, 48, 72, and 96 h) and stained with propidium iodide to analyze their cell cycle distribution by FACS (BD FACSCalibur™). Results are presented as the mean ± SEM of three independent experiments.

**Figure 4 marinedrugs-18-00211-f004:**
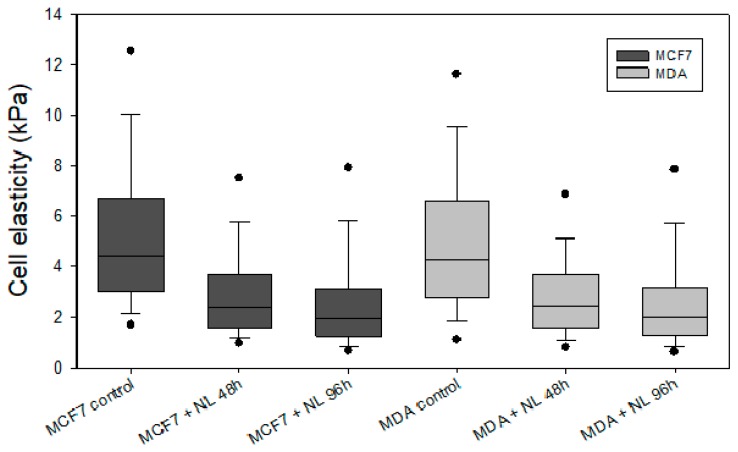
Cell stiffness of MCF-7 and MDA-MB231 cells before and after the addition of liposomes. Cells were treated with 100 µL of liposomes for the indicated time (0, 48, and 96 h), and their nanomechanical properties were measured by AFM using nanoindentation technique. Results are presented as the mean ± SEM of the values reported in Table 2.

**Table 1 marinedrugs-18-00211-t001:** The main fatty acids composition of salmon lecithin determined via gas chromatography.

Fatty Acids	Lecithin (%)	SD
C14:0	1.23	0.01
C16:0	12.09	0.03
C18:0	4.11	0.04
C21:0	1.77	0.01
C23:0	1.44	0.01
SFA	20.64	-
C16:1	1.21	0.06
C17:1	1.3	0.08
C18:1n9	20.2	0.01
MUFA	22.71	-
C18:2n6	5.79	0.01
C18:3n3	4.67	0.02
C20:4n6	2.26	0.08
C20:5n3 (EPA)	9.54	0.06
C22:4n6	1.24	0.02
C22:5n3	3.03	0.06
C22:6n3 (DHA)	30.12	0.29
PUFA	56.65	-

**Table 2 marinedrugs-18-00211-t002:** Cell stiffness of MCF-7 and MDA-MB231 cells before and after the addition of liposomes. Cells were treated with 100 µL of liposomes per 1 mL of culture media for the indicated time (0, 48, and 96 h) and the nanomechanical values were extracted from AFM force-volume images of 80 µm × 80 µm area containing at least 3–4 cells (one value per location). Notice that # corresponds to the number of force curves (elasticity values) recorded by sample (per location). Results are presented as the mean ± SEM of the values derived from the modeling of the force curves.

	MCF7 Cells	# Datum	MDA-MB-231	# Datum
No treatment	5.39 ± 3.51 kPa	5745	5.06 ± 3.27 kPa	6258
After 48 h	3.06 ± 2.32 kPa	10,203	2.94 ± 2.21 kPa	5867
After 96 h	2.74 ± 2.52 kPa	8505	2.72 ± 2.41 kPa	9318

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
