# Peer review of "Effects of Bioactive Marine-Derived Liposomes on Two Human Breast Cancer Cell Lines"

_marinedrugs, 2020, doi:10.3390/md18040211_

Round 1

Reviewer 1 Report

The graphical abstract or the image below the abstract where the authors are describing the cancer and healthy cells needs to be reworked as it is not completely understandable.

The authors need to describe the innovations, novelty of the study in the introduction section

How does the study advance the field is also missing?

The state of the art in the introduction should be added.

I would recommend the authors to add the name of the statistical test in the figure legends

Author Response

  1. Reviewer: The graphical abstract or the image below the abstract where the authors are describing the cancer and healthy cells needs to be reworked as it is not completely understandable.

Response:We thank the reviewer for the comment.The graphical abstract has been reworked to make it more understandable.

  1. Reviewer: The authors need to describe the innovations, novelty of the study in the introduction section. How does the study advance the field is also missing? The state of the art in the introduction should be added.

Response:We thank the reviewer for the comment. A paragraph further explaining the significance of this study has been added to the introduction.

  1. Reviewer: I would recommend the authors to add the name of the statistical test in the figure legends.
    Response:We thank the reviewer for the comment. The statistical test has been added to the legend of the figure.

Reviewer 2 Report

The manuscript was well written and I just have one suggestion. It would be a more completed study if the authors could do the same experiments on the third human breast cancer cell line - SKBR3 which is the HER2 positive breast cancer cell line. Including the MCF7 (ER and PR positive) and MDA-MB231 (triple negative), these three cell lines are typical models for investigating human breast cancer.

Author Response

  1. Reviewer: The manuscript was well written and I just have one suggestion. It would be a more completed study if the authors could do the same experiments on the third human breast cancer cell line - SKBR3 which is the HER2 positive breast cancer cell line. Including the MCF7 (ER and PR positive) and MDA-MB231 (triple negative), these three cell lines are typical models for investigating human breast cancer.

Response:We thank the reviewer for his positive feedback, we have choose these two cell lines notably because MCF-7 cells exhibit features of differentiated mammary epithelium whereas MDA-MB 231 is a mesenchymal breast cancer model. Even if SKBR3 cells expressed high level of HER2 compare to the two other cell lines they exhibit as for MCF7, features of epithelium. Such differences are associated with strong modifications of cell morphology, cytoskeleton organization and thus mechanical properties of plasma membrane.  However this interesting suggestion could be conducted in future studies to bring some new knowledges.

Reviewer 3 Report

The manuscript descripts the potential bio-activation of marine-derived liposomes on breast cancer cells. The authors used crystal violet assay and flow cytometry to illustrate the anti-cancer cell growth effect on liposome. This topic is interesting. However, there are many flaws through whole manuscript. Several important and major requirements listed below need to be appropriately answered.   

  1. It is difficult to understand the meaning of liposome dose (10, 20, 50 or 100µl) in figure 2. In a scientific concentration we usually express with ul/ml or uM.
  2. Since the authors didn’t provide the correct concentration of liposome used in this study, it is impossible to understand if it is too high to treat 100ul of liposome in both cancer cells. It is important for the authors to provide the evidence that 100ul of liposome (authors claim) can be reached in human serum.
  3. It is also important note that liposome reduced cancer cell number was due to apoptosis enhancement or cell proliferation reduction.
  4. What is the main cell cycle protein affected by liposome addition, in G0/G1 arrest?
  5. There is no significant differences in stiffness measurement before or after liposome additions.
  6. What is the biological meaning if cancer cells loss the stiffness?

Author Response

  1. Reviewer:It is difficult to understand the meaning of liposome dose (10, 20, 50 or 100µl) in figure 2. In a scientific concentration we usually express with ul/ml or uM.Since the authors didn’t provide the correct concentration of liposome used in this study, it is impossible to understand if it is too high to treat 100ul of liposome in both cancer cells. It is important for the authors to provide the evidence that 100ul of liposome (authors claim) can be reached in human serum.

Response:We thank the reviewer for the comment. The concentrations used were 10, 20, 50 and 100 µL/mL of medium. We have corrected this in the manuscript. These nanoparticles are going to be used as targeted drug delivery for chemotherapueitcs, so the 100 µL/mL concentration will not be present in the human serum but in the tumor microenvironment. Since we did not develop the targeted delivery system yet, we don’t think that we should prove that we can attain this concentration in vivo yet.

  1. Reviewer:It is also important note that liposome reduced cancer cell number was due to apoptosis enhancement or cell proliferation reduction.

Response:We thank the reviewer for the comment. Our cell cycle analyses Fig 3 indicate an increase of cell in GO/G1 phase and no significant variations of cells in the SubG1 fraction is observed. Thus, we can conclude that liposomes reduced the proliferation rate without enhancing cell death.

  1. Reviewer:What is the main cell cycle protein affected by liposome addition, in G0/G1 arrest?

Response:We thank the reviewer for the comment. We haven’t analyzed the molecular mechanism involved in the cell cycle arrest observed but it will be interesting to conducted a such analyzis in future studies.

  1. Reviewer:There is no significant differences in stiffness measurement before or after liposome additions.

Response:We thank the reviewer for the comment. However, we disagree, since there is a significant decrease before and after liposomes addition by about a factor of 2 in cells average stiffness. We performed ANOVA one way test on our mechanical measurements. Our statistical analysis evidenced that the differences in the median values according to cells treatments are statistically significant (P = <0.001). These resultats are also confirmed by all pairwise multiple comparison procedures according to Dunn's Method (P = <0.050).

  1. Reviewer:What is the biological meaning if cancer cells loss the stiffness?

Response:We thank the reviewer for the comment. The abnormal stiffening of the tumor environment is associated to tumor development. Through a local denaturation of adjacent tissues, nanomaterials can induce a remodeling and softening of the tumor microenvironment which can lead to tumor regression. So the loss of stiffness of cancer cells represent a mechanical cue for tumor regression and enhance the delivery and thus the

Round 2

Reviewer 3 Report

1: The authors claimed “These nanoparticles are going to be used as targeted drug delivery for chemotherapueitcs, so the 100 µL/mL concentration will not be present in the human serum but in the tumor microenvironment” without any evidence or prove. I don’t think it is an appropriate way to response the comment.  

2: Apoptosis detection such as PARP or Caspase cleavage in western blot, annexin v analysis in flow cytometer must be performed.

3: Some cell cycle arrest proteins in G1 phase must be analyzed.

4: Cell proliferation of BrdU incorporation must to be performed.

5: No any significance was noticed in figure 4 or table 2. According to the date presents in figure 4. It is very hard to believe any significance exit in before and after liposome treatments. It will be better to provide the original data in supplementary file.

6: Since liposome induced the lost of stiffness, the authors also explained the lost o stiffness could induced tumor remodeling, it should be easy to see more migration and invasion abilities after liposome treatment on cancer cells. This is the essential experiments.